# Expectation-Maximization
# for Learning Determinantal Point Processes

**Jennifer Gillenwater**
Computer and Information Science
University of Pennsylvania
jengi@cis.upenn.edu

**Alex Kulesza**
Computer Science and Engineering
University of Michigan
kulesza@umich.edu

**Emily Fox**
Statistics
University of Washington
ebfox@stat.washington.edu

**Ben Taskar**
Computer Science and Engineering
University of Washington
taskar@cs.washington.edu

## Abstract

A determinantal point process (DPP) is a probabilistic model of set diversity compactly parameterized by a positive semi-definite kernel matrix. To fit a DPP to a given task, we would like to learn the entries of its kernel matrix by maximizing the log-likelihood of the available data. However, log-likelihood is non-convex in the entries of the kernel matrix, and this learning problem is conjectured to be NP-hard [1]. Thus, previous work has instead focused on more restricted convex learning settings: learning only a single weight for each row of the kernel matrix [2], or learning weights for a linear combination of DPPs with fixed kernel matrices [3]. In this work we propose a novel algorithm for learning the full kernel matrix. By changing the kernel parameterization from matrix entries to eigenvalues and eigenvectors, and then lower-bounding the likelihood in the manner of expectation-maximization algorithms, we obtain an effective optimization procedure. We test our method on a real-world product recommendation task, and achieve relative gains of up to 16.5% in test log-likelihood compared to the naive approach of maximizing likelihood by projected gradient ascent on the entries of the kernel matrix.

## 1 Introduction

Subset selection is a core task in many real-world applications. For example, in product recommendation we typically want to choose a small set of products from a large collection; many other examples of subset selection tasks turn up in domains like document summarization [4, 5], sensor placement [6, 7], image search [3, 8], and auction revenue maximization [9], to name a few. In these applications, a good subset is often one whose individual items are all high-quality, but also all distinct. For instance, recommended products should be popular, but they should also be diverse to increase the chance that a user finds at least one of them interesting. Determinantal point processes (DPPs) offer one way to model this tradeoff; a DPP defines a distribution over all possible subsets of a ground set, and the mass it assigns to any given set is a balanced measure of that set's quality and diversity.

Originally discovered as models of fermions [10], DPPs have recently been effectively adapted for a variety of machine learning tasks [8, 11, 12, 13, 14, 15, 16, 17, 18, 19, 2, 3, 20]. They offer attractive computational properties, including exact and efficient normalization, marginalization, conditioning, and sampling [21]. These properties arise in part from the fact that a DPP can be compactly param-

eterized by an $N \times N$ positive semi-definite matrix $L$. Unfortunately, though, learning $L$ from example subsets by maximizing likelihood is conjectured to be NP-hard [1, Conjecture 4.1]. While gradient ascent can be applied in an attempt to approximately optimize the likelihood objective, we show later that it requires a projection step that often produces degenerate results.

For this reason, in most previous work only partial learning of $L$ has been attempted. [2] showed that the problem of learning a scalar weight for each row of $L$ is a convex optimization problem. This amounts to learning what makes an item high-quality, but does not address the issue of what makes two items similar. [3] explored a different direction, learning weights for a linear combination of DPPs with fixed $L$s. This works well in a limited setting, but requires storing a potentially large set of kernel matrices, and the final distribution is no longer a DPP, which means that many attractive computational properties are lost. [8] proposed as an alternative that one first assume $L$ takes on a particular parametric form, and then sample from the posterior distribution over kernel parameters using Bayesian methods. This overcomes some of the disadvantages of [3]'s $L$-ensemble method, but does not allow for learning an unconstrained, non-parametric $L$.

The learning method we propose in this paper differs from those of prior work in that it does not assume fixed values or restrictive parameterizations for $L$, and exploits the eigendecomposition of $L$. Many properties of a DPP can be simply characterized in terms of the eigenvalues and eigenvectors of $L$, and working with this decomposition allows us to develop an expectation-maximization (EM) style optimization algorithm. This algorithm negates the need for the problematic projection step that is required for naive gradient ascent to maintain positive semi-definiteness of $L$. As the experiments show, a projection step can sometimes lead to learning a nearly diagonal $L$, which fails to model the negative interactions between items. These interactions are vital, as they lead to the diversity-seeking nature of a DPP. The proposed EM algorithm overcomes this failing, making it more robust to initialization and dataset changes. It is also asymptotically faster than gradient ascent.

## 2   Background

Formally, a DPP $\mathcal{P}$ on a ground set of items $\mathcal{Y} = \{1, \ldots, N\}$ is a probability measure on $2^{\mathcal{Y}}$, the set of all subsets of $\mathcal{Y}$. For every $Y \subseteq \mathcal{Y}$ we have $\mathcal{P}(Y) \propto \det(L_Y)$, where $L$ is a positive semi-definite (PSD) matrix. The subscript $L_Y \equiv [L_{ij}]_{i,j \in Y}$ denotes the restriction of $L$ to the entries indexed by elements of $Y$, and we have $\det(L_\emptyset) \equiv 1$. Notice that the restriction to PSD matrices ensures that all principal minors of $L$ are non-negative, so that $\det(L_Y) \geq 0$ as required for a proper probability distribution. The normalization constant for the distribution can be computed explicitly thanks to the fact that $\sum_Y \det(L_Y) = \det(L + I)$, where $I$ is the $N \times N$ identity matrix. Intuitively, we can think of a diagonal entry $L_{ii}$ as capturing the quality of item $i$, while an off-diagonal entry $L_{ij}$ measures the similarity between items $i$ and $j$.

An alternative representation of a DPP is given by the *marginal kernel*: $K = L(L + I)^{-1}$. The $L$-$K$ relationship can also be written in terms of their eigendecompositons. $L$ and $K$ share the same eigenvectors $\boldsymbol{v}$, and an eigenvalue $\lambda_i$ of $K$ corresponds to an eigenvalue $\lambda_i/(1 - \lambda_i)$ of $L$:

$$K = \sum_{j=1}^{N} \lambda_j \boldsymbol{v}_j \boldsymbol{v}_j^\top \quad \Leftrightarrow \quad L = \sum_{j=1}^{N} \frac{\lambda_j}{1 - \lambda_j} \boldsymbol{v}_j \boldsymbol{v}_j^\top \ . \tag{1}$$

Clearly, if $L$ if PSD then $K$ is as well, and the above equations also imply that the eigenvalues of $K$ are further restricted to be $\leq 1$. $K$ is called the *marginal* kernel because, for any set $Y \sim \mathcal{P}$ and for every $A \subseteq \mathcal{Y}$:

$$\mathcal{P}(A \subseteq Y) = \det(K_A) \,. \tag{2}$$

We can also write the exact (non-marginal, normalized) probability of a set $Y \sim \mathcal{P}$ in terms of $K$:

$$\mathcal{P}(Y) = \frac{\det(L_Y)}{\det(L + I)} = |\det(K - I_{\overline{Y}})| \,, \tag{3}$$

where $I_{\overline{Y}}$ is the identity matrix with entry $(i, i)$ zeroed for items $i \in Y$ [1, Equation 3.69]. In what follows we use the $K$-based formula for $\mathcal{P}(Y)$ and learn the marginal kernel $K$. This is equivalent to learning $L$, as Equation (1) can be applied to convert from $K$ to $L$.

# 3 Learning algorithms

In our learning setting the input consists of $n$ example subsets, $\{Y_1, \ldots, Y_n\}$, where $Y_i \subseteq \{1, \ldots, N\}$ for all $i$. Our goal is to maximize the likelihood of these example sets. We first describe in Section 3.1 a naive optimization procedure: projected gradient ascent on the entries of the marginal matrix $K$, which will serve as a baseline in our experiments. We then develop an EM method: Section 3.2 changes variables from kernel entries to eigenvalues and eigenvectors (introducing a hidden variable in the process), Section 3.3 applies Jensen's inequality to lower-bound the objective, and Sections 3.4 and 3.5 outline a coordinate ascent procedure on this lower bound.

## 3.1 Projected gradient ascent

The log-likelihood maximization problem, based on Equation (3), is:

$$\max_K \sum_{i=1}^n \log\left(|\det(K - I_{\overline{Y}_i})|\right) \quad \text{s.t. } K \succeq 0, \ I - K \succeq 0 \tag{4}$$

where the first constraint ensures that $K$ is PSD and the second puts an upper limit of $1$ on its eigenvalues. Let $\mathcal{L}(K)$ represent this log-likelihood objective. Its partial derivative with respect to $K$ is easy to compute by applying a standard matrix derivative rule [22, Equation 57]:

$$\frac{\partial \mathcal{L}(K)}{\partial K} = \sum_{i=1}^n (K - I_{\overline{Y}_i})^{-1}. \tag{5}$$

Thus, projected gradient ascent [23] is a viable, simple optimization technique. Algorithm 1 outlines this method, which we refer to as K-Ascent (KA). The initial $K$ supplied as input to the algorithm can be any PSD matrix with eigenvalues $\leq 1$. The first part of the projection step, $\max(\boldsymbol{\lambda}, 0)$, chooses the closest (in Frobenius norm) PSD matrix to $Q$ [24, Equation 1]. The second part, $\min(\boldsymbol{\lambda}, 1)$, caps the eigenvalues at $1$. (Notice that only the eigenvalues have to be projected; $K$ remains symmetric after the gradient step, so its eigenvectors are already guaranteed to be real.)

Unfortunately, the projection can take us to a poor local optima. To see this, consider the case where the starting kernel $K$ is a poor fit to the data. In this case, a large initial step size $\eta$ will probably be accepted; even though such a step will likely result in the truncation of many eigenvalues at $0$, the resulting matrix will still be an improvement over the poor initial $K$. However, with many zero eigenvalues, the new $K$ will be near-diagonal, and, unfortunately, Equation (5) dictates that if the current $K$ is diagonal, then its gradient is as well. Thus, the KA algorithm cannot easily move to any highly non-diagonal matrix. It is possible that employing more complex step-size selection mechanisms could alleviate this problem, but the EM algorithm we develop in the next section will negate the need for these entirely.

The EM algorithm we develop also has an advantage in terms of asymptotic runtime. The computational complexity of KA is dominated by the matrix inverses of the $\mathcal{L}$ derivative, each of which requires $O(N^3)$ operations, and by the eigendecomposition needed for the projection, also $O(N^3)$. The overall runtime of KA, assuming $T_1$ iterations until convergence and an average of $T_2$ iterations to find a step size, is $O(T_1 n N^3 + T_1 T_2 N^3)$. As we will show in the following sections, the overall runtime of the EM algorithm is $O(T_1 n N k^2 + T_1 T_2 N^3)$, which can be substantially better than KA's runtime for $k \ll N$.

## 3.2 Eigendecomposing

Eigendecomposition is key to many core DPP algorithms such as sampling and marginalization. This is because the eigendecomposition provides an alternative view of the DPP as a generative process, which often leads to more efficient algorithms. Specifically, sampling a set $Y$ can be broken down into a two-step process, the first of which involves generating a hidden variable $J \subseteq \{1, \ldots, N\}$ that codes for a particular set of $K$'s eigenvectors. We review this process below, then exploit it to develop an EM optimization scheme.

Suppose $K = V \Lambda V^\top$ is an eigendecomposition of $K$. Let $V^J$ denote the submatrix of $V$ containing only the columns corresponding to the indices in a set $J \subseteq \{1, \ldots, N\}$. Consider the corresponding

| **Algorithm 1** K-Ascent (KA) | **Algorithm 2** Expectation-Maximization (EM) |
|---|---|
| **Input:** $K, \{Y_1, \ldots, Y_n\}, c$ | **Input:** $K, \{Y_1, \ldots, Y_n\}, c$ |
| **repeat** | Eigendecompose $K$ into $V, \boldsymbol{\lambda}$ |
| $\quad G \leftarrow \frac{\partial \mathcal{L}(K)}{\partial K}$ (Eq. 5) | **repeat** |
| $\quad \eta \leftarrow 1$ | $\quad$ **for** $j = 1, \ldots, N$ **do** |
| $\quad$ **repeat** | $\quad\quad \boldsymbol{\lambda}'_j \leftarrow \frac{1}{n} \sum_i p_K(j \in J \mid Y_i)$ (Eq. 19) |
| $\quad\quad Q \leftarrow K + \eta G$ | $\quad G \leftarrow \frac{\partial F(V, \boldsymbol{\lambda}')}{\partial V}$ (Eq. 20) |
| $\quad\quad$ Eigendecompose $Q$ into $V, \boldsymbol{\lambda}$ | $\quad \eta \leftarrow 1$ |
| $\quad\quad \boldsymbol{\lambda} \leftarrow \min(\max(\boldsymbol{\lambda}, 0), 1)$ | $\quad$ **repeat** |
| $\quad\quad Q \leftarrow V \mathrm{diag}(\boldsymbol{\lambda}) V^\top$ | $\quad\quad V' \leftarrow V \exp[\eta \left(V^\top G - G^\top V\right)]$ |
| $\quad\quad \eta \leftarrow \frac{\eta}{2}$ | $\quad\quad \eta \leftarrow \frac{\eta}{2}$ |
| $\quad$ **until** $\mathcal{L}(Q) > \mathcal{L}(K)$ | $\quad$ **until** $\mathcal{L}(V', \boldsymbol{\lambda}') > \mathcal{L}(V, \boldsymbol{\lambda}')$ |
| $\quad \delta \leftarrow \mathcal{L}(Q) - \mathcal{L}(K)$ | $\quad \delta \leftarrow F(V', \boldsymbol{\lambda}') - F(V, \boldsymbol{\lambda})$ |
| $\quad K \leftarrow Q$ | $\quad \boldsymbol{\lambda} \leftarrow \boldsymbol{\lambda}', \ \ V \leftarrow V', \ \ \eta \leftarrow 2\eta$ |
| **until** $\delta < c$ | **until** $\delta < c$ |
| **Output:** $K$ | **Output:** $K$ |

marginal kernel, with all selected eigenvalues set to 1:

$$K^{V^J} = \sum_{j \in J} \boldsymbol{v}_j \boldsymbol{v}_j^\top = V^J (V^J)^\top. \tag{6}$$

Any such kernel whose eigenvalues are all 1 is called an *elementary* DPP. According to [21, Theorem 7], a DPP with marginal kernel $K$ is a mixture of all $2^N$ possible elementary DPPs:

$$\mathcal{P}(Y) = \sum_{J \subseteq \{1, \ldots, N\}} \mathcal{P}^{V^J}(Y) \prod_{j \in J} \lambda_j \prod_{j \notin J} (1 - \lambda_j), \qquad \mathcal{P}^{V^J}(Y) = \mathbf{1}(|Y| = |J|) \det(K_Y^{V^J}). \tag{7}$$

This perspective leads to an efficient DPP sampling algorithm, where a set $J$ is first chosen according to its mixture weight in Equation (7), and then a simple algorithm is used to sample from $P^{V^J}$ [5, Algorithm 1]. In this sense, the index set $J$ is an intermediate hidden variable in the process for generating a sample $Y$.

We can exploit this hidden variable $J$ to develop an EM algorithm for learning $K$. Re-writing the data log-likelihood to make the hidden variable explicit:

$$\mathcal{L}(K) = \mathcal{L}(\Lambda, V) = \sum_{i=1}^{n} \log \left( \sum_J p_K(J, Y_i) \right) = \sum_{i=1}^{n} \log \left( \sum_J p_K(Y_i \mid J) p_K(J) \right), \quad \text{where} \tag{8}$$

$$p_K(J) = \prod_{j \in J} \lambda_j \prod_{j \notin J} (1 - \lambda_j), \qquad p_K(Y_i \mid J) = \mathbf{1}(|Y_i| = |J|) \det([V^J (V^J)^\top]_{Y_i}). \tag{9}$$

These equations follow directly from Equations (6) and (7).

### 3.3 Lower bounding the objective

We now introduce an auxiliary distribution, $q(J \mid Y_i)$, and deploy it with Jensen's inequality to lower-bound the likelihood objective. This is a standard technique for developing EM schemes for dealing with hidden variables [25]. Proceeding in this direction:

$$\mathcal{L}(V, \Lambda) = \sum_{i=1}^{n} \log \left( \sum_J q(J \mid Y_i) \frac{p_K(J, Y_i)}{q(J \mid Y_i)} \right) \geq \sum_{i=1}^{n} \sum_J q(J \mid Y_i) \log \left( \frac{p_K(J, Y_i)}{q(J \mid Y_i)} \right) \equiv F(q, V, \Lambda). \tag{10}$$

The function $F(q, V, \Lambda)$ can be expressed in either of the following two forms:

$$F(q, V, \Lambda) = \sum_{i=1}^{n} -\mathbf{KL}(q(J \mid Y_i) \parallel p_K(J \mid Y_i)) + \mathcal{L}(V, \Lambda) \tag{11}$$

$$= \sum_{i=1}^{n} \mathbb{E}_q[\log p_K(J, Y_i)] + H(q) \tag{12}$$

where $H$ is entropy. Consider optimizing this new objective by coordinate ascent. From Equation (11) it is clear that, holding $V, \Lambda$ constant, $F$ is concave in $q$. This follows from the concavity of $\mathbf{KL}$ divergence. Holding $q$ constant in Equation (12) yields the following function:

$$F(V, \Lambda) = \sum_{i=1}^{n} \sum_{J} q(J \mid Y_i) \left[ \log p_K(J) + \log p_K(Y_i \mid J) \right]. \tag{13}$$

This expression is concave in $\lambda_j$, since $\log$ is concave. However, it is not concave in $V$ due to the non-convex $V^\top V = I$ constraint. We describe in Section 3.5 one way to handle this.

To summarize, coordinate ascent on $F(q, V, \Lambda)$ alternates the following "expectation" and "maximization" steps; the first is concave in $q$, and the second is concave in the eigenvalues:

$$\text{E-step:} \quad \min_q \sum_{i=1}^{n} \mathbf{KL}(q(J \mid Y_i) \parallel p_K(J \mid Y_i)) \tag{14}$$

$$\text{M-step:} \quad \max_{V, \Lambda} \sum_{i=1}^{n} \mathbb{E}_q[\log p_K(J, Y_i)] \ \text{s.t.} \ \mathbf{0} \le \boldsymbol{\lambda} \le \mathbf{1}, V^\top V = I \tag{15}$$

### 3.4 E-step

The E-step is easily solved by setting $q(J \mid Y_i) = p_K(J \mid Y_i)$, which minimizes the KL divergence. Interestingly, we can show that this distribution is itself a conditional DPP, and hence can be compactly described by an $N \times N$ kernel matrix. Thus, to complete the E-step, we simply need to construct this kernel. Lemma 1 (see the supplement for a proof) gives an explicit formula. Note that $q$'s probability mass is restricted to sets of a particular size $k$, and hence we call it a $k$-DPP. A $k$-DPP is a variant of DPP that can also be efficiently sampled from and marginalized, via modifications of the standard DPP algorithms. (See the supplement and [3] for more on $k$-DPPs.)

**Lemma 1.** *At the completion of the E-step, $q(J \mid Y_i)$ with $|Y_i| = k$ is a $k$-DPP with (non-marginal) kernel $Q^{Y_i}$:*

$$Q^{Y_i} = R Z^{Y_i} R, \ \text{and} \ q(J \mid Y_i) \propto \mathbf{1}(|Y_i| = |J|) \det(Q_J^{Y_i}), \ \text{where} \tag{16}$$

$$U = V^\top, \quad Z^{Y_i} = U^{Y_i}(U^{Y_i})^\top, \ \text{and} \ R = \text{diag}\left(\sqrt{\boldsymbol{\lambda}/(\mathbf{1} - \boldsymbol{\lambda})}\right). \tag{17}$$

### 3.5 M-step

The M-step update for the eigenvalues is a closed-form expression with no need for projection. Taking the derivative of Equation (13) with respect to $\lambda_j$, setting it equal to zero, and solving for $\lambda_j$:

$$\lambda_j = \frac{1}{n} \sum_{i=1}^{n} \sum_{J:j \in J} q(J \mid Y_i). \tag{18}$$

The exponential-sized sum here is impractical, but we can eliminate it. Recall from Lemma 1 that $q(J \mid Y_i)$ is a $k$-DPP with kernel $Q^{Y_i}$. Thus, we can use $k$-DPP marginalization algorithms to efficiently compute the sum over $J$. More concretely, let $\hat{V}$ represent the eigenvectors of $Q^{Y_i}$, with $\hat{v}_r(j)$ indicating the $j$th element of the $r$th eigenvector. Then the marginals are:

$$\sum_{J:j \in J} q(J \mid Y_i) = q(j \in J \mid Y_i) = \sum_{r=1}^{N} \hat{v}_r(j)^2, \tag{19}$$

which allows us to compute the eigenvalue updates in time $O(nNk^2)$, for $k = \max_i |Y_i|$. (See the supplement for the derivation of Equation (19) and its computational complexity.) Note that this update is self-normalizing, so explicit enforcement of the $0 \leq \lambda_j \leq 1$ constraint is unnecessary. There is one small caveat: the $Q^{Y_i}$ matrix will be infinite if any $\lambda_j$ is exactly equal to 1 (due to $R$ in Equation (17)). In practice, we simply tighten the constraint on $\boldsymbol{\lambda}$ to keep it slightly below 1.

Turning now to the M-step update for the eigenvectors, the derivative of Equation (13) with respect to $V$ involves an exponential-size sum over $J$ similar to that of the eigenvalue derivative. However, the terms of the sum in this case depend on $V$ as well as on $q(J \mid Y_i)$, making it hard to simplify. Yet, for the particular case of the initial gradient, where we have $q = p$, simplification is possible:

$$\frac{\partial F(V, \Lambda)}{\partial V} = \sum_{i=1}^{n} 2B_{Y_i}(H^{Y_i})^{-1}V_{Y_i}R^2 \qquad (20)$$

where $H^{Y_i}$ is the $|Y_i| \times |Y_i|$ matrix $V_{Y_i}R^2V_{Y_i}^\top$ and $V_{Y_i} = (U^{Y_i})^\top$. $B_{Y_i}$ is a $N \times |Y_i|$ matrix containing the columns of the $N \times N$ identity corresponding to items in $Y_i$; $B_{Y_i}$ simply serves to map the gradients with respect to $V_{Y_i}$ into the proper positions in $V$. This formula allows us to compute the eigenvector derivatives in time $O(nNk^2)$, where again $k = \max_i |Y_i|$. (See the supplement for the derivation of Equation (20) and its computational complexity.)

Equation (20) is only valid for the first gradient step, so in practice we do not bother to fully optimize $V$ in each M-step; we simply take a single gradient step on $V$. Ideally we would repeatedly evaluate the M-step objective, Equation (13), with various step sizes to find the optimal one. However, the M-step objective is intractable to evaluate exactly, as it is an expectation with respect to an exponential-size distribution. In practice, we solve this issue by performing an E-step for each trial step size. That is, we update $q$'s distribution to match the updated $V$ and $\Lambda$ that define $p_K$, and then determine if the current step size is good by checking for improvement in the likelihood $\mathcal{L}$.

There is also the issue of enforcing the non-convex constraint $V^\top V = I$. We could project $V$ to ensure this constraint, but, as previously discussed for eigen*values*, projection steps often lead to poor local optima. Thankfully, for the particular constraint associated with $V$, more sophisticated update techniques exist: the constraint $V^\top V = I$ corresponds to optimization over a Stiefel manifold, so the algorithm from [26, Page 326] can be employed. In practice, we simplify this algorithm by neglecting second-order information (the Hessian) and using the fact that the $V$ in our application is full-rank. With these simplifications, the following multiplicative update is all that is needed:

$$V \leftarrow V \exp\left[\eta\left(V^\top \frac{\partial \mathcal{L}}{\partial V} - \left(\frac{\partial \mathcal{L}}{\partial V}\right)^\top V\right)\right], \qquad (21)$$

where $\exp$ denotes the matrix exponential and $\eta$ is the step size. Algorithm 2 summarizes the overall EM method. As previously mentioned, assuming $T_1$ iterations until convergence and an average of $T_2$ iterations to find a step size, its overall runtime is $O(T_1 nNk^2 + T_1 T_2 N^3)$. The first term in this complexity comes from the eigenvalue updates, Equation (19), and the eigenvector derivative computation, Equation (20). The second term comes from repeatedly computing the Stiefel manifold update of $V$, Equation (21), during the step size search.

## 4  Experiments

We test the proposed EM learning method (Algorithm 2) by comparing it to K-Ascent (KA, Algorithm 1)[1]. Both methods require a starting marginal kernel $\tilde{K}$. Note that neither EM nor KA can deal well with starting from a kernel with too many zeros. For example, starting from a diagonal kernel, both gradients, Equations (5) and (20), will be diagonal, resulting in no modeling of diversity. Thus, the two initialization options that we explore have non-trivial off-diagonals. The first of these options is relatively naive, while the other incorporates statistics from the data.

For the first initialization type, we use a Wishart distribution with $N$ degrees of freedom and an identity covariance matrix to draw $\tilde{L} \sim \mathcal{W}_N(N, I)$. The Wishart distribution is relatively unassuming: in terms of eigenvectors, it spreads its mass uniformly over all unitary matrices [27]. We make

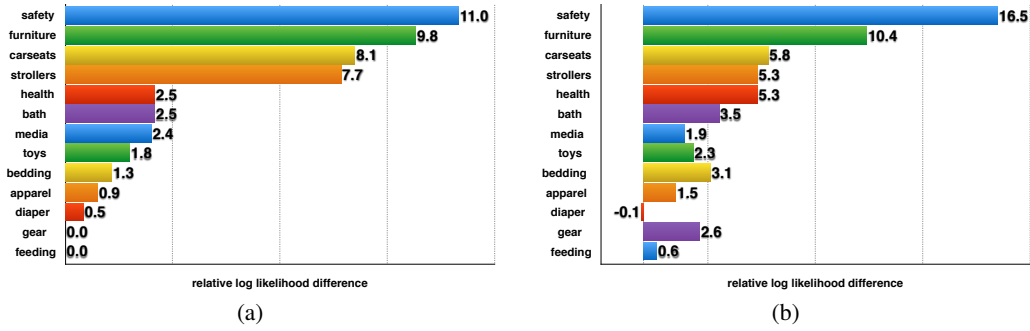

Figure 1: Relative test log-likelihood differences, $100\frac{(\text{EM}-\text{KA})}{|\text{KA}|}$, using: (a) Wishart initialization in the full-data setting, and (b) moments-matching initialization in the data-poor setting.

just one simple modification to its output to make it a better fit for practical data: we re-scale the resulting matrix by $1/N$ so that the corresponding DPP will place a non-trivial amount of probability mass on small sets. (The Wishart's mean is $NI$, so it tends to over-emphasize larger sets unless we re-scale.) We then convert $\tilde{L}$ to $\tilde{K}$ via Equation (1).

For the second initialization type, we employ a form of moment matching. Let $m_i$ and $m_{ij}$ represent the normalized frequencies of single items and pairs of items in the training data:

$$m_i = \frac{1}{n}\sum_{\ell=1}^{n}\mathbf{1}(i \in Y_\ell), \quad m_{ij} = \frac{1}{n}\sum_{\ell=1}^{n}\mathbf{1}(i \in Y_\ell \wedge j \in Y_\ell). \tag{22}$$

Recalling Equation (2), we attempt to match the first and second order moments by choosing $\tilde{K}$ as:

$$\tilde{K}_{ii} = m_i, \quad \tilde{K}_{ij} = \sqrt{\max\left(\tilde{K}_{ii}\tilde{K}_{jj} - m_{ij}, 0\right)}. \tag{23}$$

To ensure a valid starting kernel, we then project $\tilde{K}$ by clipping its eigenvalues at 0 and 1.

### 4.1 Baby registry tests

Consider a product recommendation task, where the ground set comprises $N$ products that can be added to a particular category (e.g., toys or safety) in a baby registry. A very simple recommendation system might suggest products that are popular with other consumers; however, this does not account for negative interactions: if a consumer has already chosen a carseat, they most likely will not choose an additional carseat, no matter how popular it is with other consumers. DPPs are ideal for capturing such negative interactions. A learned DPP could be used to populate an initial, basic registry, as well as to provide live updates of product recommendations as a consumer builds their registry.

To test our DPP learning algorithms, we collected a dataset consisting of 29,632 baby registries from Amazon.com, filtering out those listing fewer than 5 or more than 100 products. Amazon characterizes each product in a baby registry as belonging to one of 18 categories, such as "toys" and "safety". For each registry, we created sub-registries by splitting it according to these categories. (A registry with 5 toy items and 10 safety items produces two sub-registries.) For each category, we then filtered down to its top 100 most frequent items, and removed any product that did not occur in at least 100 sub-registries. We discarded categories with $N < 25$ or fewer than $2N$ remaining (non-empty) sub-registries for training. The resulting 13 categories have an average inventory size of $N = 71$ products and an average number of sub-registries $n = 8,585$. We used 70% of the data for training and 30% for testing. Note that categories such as "carseats" contain more diverse items than just their namesake; for instance, "carseats" also contains items such as seat back kick protectors and rear-facing baby view mirrors. See the supplement for more dataset details and for quartile numbers for all of the experiments.

Figure 1a shows the relative test log-likelihood differences of EM and KA when starting from a Wishart initialization. These numbers are the medians from 25 trials (draws from the Wishart). EM

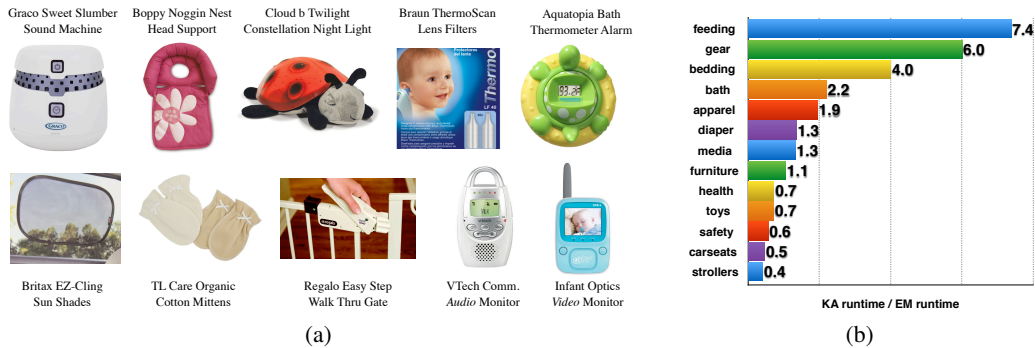

Figure 2: (a) A high-probability set of size $k = 10$ selected using an EM model for the "safety" category. (b) Runtime ratios.

gains an average of 3.7%, but has a much greater advantage for some categories than for others. Speculating that EM has more of an advantage when the off-diagonal components of $K$ are truly important—when products exhibit strong negative interactions—we created a matrix $M$ for each category with the true data marginals from Equation (22) as its entries. We then checked the value of $d = \frac{1}{N} \frac{||M||_F}{||\text{diag}(M)||_2}$. This value correlates well with the relative gains for EM: the 4 categories for which EM has the largest gains (safety, furniture, carseats, and strollers) all exhibit $d > 0.025$, while categories such as feeding and gear have $d < 0.012$. Investigating further, we found that, as foreshadowed in Section 3.1, KA performs particularly poorly in the high-$d$ setting because of its projection step—projection can result in KA learning a near-diagonal matrix.

If instead of the Wishart initialization we use the moments-matching initializer, this alleviates KA's projection problem, as it provides a starting point closer to the true kernel. With this initializer, KA and EM have comparable test log-likelihoods (average EM gain of 0.4%). However, the moments-matching initializer is not a perfect fix for the KA algorithm in all settings. For instance, consider a data-poor setting, where for each category we have only $n = 2N$ training examples. In this case, even with the moments-matching initializer EM has a significant edge over KA, as shown in Figure 1b: EM gains an average of 4.5%, with a maximum gain of 16.5% for the safety category.

To give a concrete example of the advantages of EM training, Figure 2a shows a greedy approximation [28, Section 4] to the most-likely ten-item registry in the category "safety", according to a Wishart-initialized EM model. The corresponding KA selection differs from Figure 2a in that it replaces the lens filters and the head support with two additional baby monitors: "Motorola MBP36 Remote Wireless Video Baby Monitor", and "Summer Infant Baby Touch Digital Color Video Monitor". It seems unlikely that many consumers would select three different brands of video monitor.

Having established that EM is more robust than KA, we conclude with an analysis of runtimes. Figure 2b shows the ratio of KA's runtime to EM's for each category. As discussed earlier, EM is asymptotically faster than KA, and we see this borne out in practice even for the moderate values of $N$ and $n$ that occur in our registries dataset: on average, EM is 2.1 times faster than KA.

## 5    Conclusion

We have explored learning DPPs in a setting where the kernel $K$ is not assumed to have fixed values or a restrictive parametric form. By exploiting $K$'s eigendecomposition, we were able to develop a novel EM learning algorithm. On a product recommendation task, we have shown EM to be faster and more robust than the naive approach of maximizing likelihood by projected gradient. In other applications for which modeling negative interactions between items is important, we anticipate that EM will similarly have a significant advantage.

### Acknowledgments

This work was supported in part by ONR Grant N00014-10-1-0746.

## Footnotes

[1]Code and data for all experiments can be downloaded from https://code.google.com/p/em-for-dpps

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
