[Supplementary Material]

# A Proof of Lemma 1

Lemma 1 gives the exact form of $q$'s kernel. Before giving the proof, we briefly note that $q$ differs slightly from the typical DPPs we have seen thus far, in its *conditional* nature. More precisely, for a set $Y_i$ of size $k$, $q$ qualifies as a *$k$-DPP*, a DPP conditioned on sets of size $k$. Formally, a $k$-DPP with (non-marginal) kernel $L$ assigns probability $\propto \det(L_Y)$ for $|Y| = k$, and probability zero for $|Y| \neq k$. As for regular DPPs, a $k$-DPP can be efficiently sampled from and marginalized, via modifications of the standard DPP algorithms. For example, the normalization constant for a $k$-DPP is given by the identity $\sum_{Y:|Y|=k} \det(L_Y) = e_k^N(L)$, where $e_k^N(L)$ represents the $k$th-order elementary symmetric polynomial on the eigenvalues of $L$ [1]. [2]'s "summation algorithm" computes $e_k^N(L)$ in $O(Nk)$ time. In short $k$-DPPs enjoy many of the advantages of DPPs. Their identical parameterization, in terms of a single kernel, makes our E-step simple, and their normalization and marginalization properties are useful for the M-step updates.

**Lemma 1.** *At the completion of the E-step, $q(J \mid Y_i)$ with $|Y_i| = k$ is a $k$-DPP with (non-marginal) kernel $Q^{Y_i}$:*

$$Q^{Y_i} = RZ^{Y_i}R, \quad \text{and} \quad q(J \mid Y_i) \propto \mathbf{1}(|Y_i| = |J|)\det(Q_J^{Y_i}), \text{ where} \tag{A.1}$$

$$U = V^\top, \quad Z^{Y_i} = U^{Y_i}(U^{Y_i})^\top, \quad \text{and} \quad R = \text{diag}\left(\sqrt{\boldsymbol{\lambda}/(\mathbf{1} - \boldsymbol{\lambda})}\right). \tag{A.2}$$

*Proof.* Since the E-step is an unconstrained KL divergence minimization, we have:

$$q(J \mid Y_i) = p_K(J \mid Y_i) = \frac{p_K(J, Y_i)}{p_K(Y_i)} \propto p_K(J, Y_i) = p_K(J)p_K(Y_i \mid J) \tag{A.3}$$

where the proportionality follows because $Y_i$ is held constant in the conditional $q$ distribution. Recalling Equation (9), notice that $p_K(Y_i \mid J)$ can be re-expressed as follows:

$$p_K(Y_i \mid J) = \mathbf{1}(|Y_i| = |J|)\det([V^J(V^J)^\top]_{Y_i}) = \mathbf{1}(|Y_i| = |J|)\det([U^{Y_i}(U^{Y_i})^\top]_J). \tag{A.4}$$

This follows from the identity $\det(AA^\top) = \det(A^\top A)$, for any full-rank square matrix $A$. The subsequent swapping of $J$ and $Y_i$, once $V^\top$ is re-written as $U$, does not change the indexed submatrix.

Plugging this back into Equation (A.3):

$$q(J \mid Y_i) \propto p_K(J)\mathbf{1}(|Y_i| = |J|)\det([U^{Y_i}(U^{Y_i})^\top]_J) = p_K(J)\mathbf{1}(|Y_i| = |J|)P^{U^{Y_i}}(J) \tag{A.5}$$

where $P^{U^{Y_i}}$ represents an elementary DPP, just as in Equation (6), but over $J$ rather than $Y$. Multiplying this expression by a term that is constant for all $J$ maintains proportionality and allows us to simplify the the $p_K(J)$ term. Taking the definition of $p_K(J)$ from Equation (9):

$$q(J \mid Y_i) \propto \left(\prod_{j=1}^N \frac{1}{1-\lambda_j}\right)\mathbf{1}(|Y_i| = |J|)P^{U^{Y_i}}(J)\prod_{j\in J}\lambda_j\prod_{j\notin J}(1-\lambda_j) \tag{A.6}$$

$$= \mathbf{1}(|Y_i| = |J|)P^{U^{Y_i}}(J)\prod_{j\in J}\frac{\lambda_j}{1-\lambda_j} \tag{A.7}$$

Having eliminated all dependence on $j \notin J$, it is now possible to express $q(J \mid Y_i)$ as the $J$ principal minor of a PSD kernel matrix (see $Q^{Y_i}$ in the statement of the lemma). Thus, $q$ is a $k$-DPP. $\qquad\square$

# B  M-Step eigenvalue updates

We can exploit standard $k$-DPP marginalization formulas to efficiently compute the eigenvalue updates for EM. Specifically, the exponential-size sum over $J$ from Equation (18) can be reduced to the computation of an eigendecomposition and several elementary symmetric polynomials on the resulting eigenvalues. Let $e_{k-1}^{-j}(Q^{Y_i})$ be the $(k-1)$-order elementary symmetric polynomial over all eigenvalues of $Q^{Y_i}$ except for the $j$th one. Then, by direct application of [3, Equation 205], $q$'s singleton marginals are:

$$\sum_{J:j\in J} q(J \mid Y_i) = q(j \in J \mid Y_i) = \frac{1}{e_{|Y_i|}^N(Q^{Y_i})} \sum_{r=1}^N \hat{v}_r(j)^2 \hat{\lambda}_r e_{|Y_i|-1}^{-r}(Q^{Y_i}). \tag{B.8}$$

As previously noted, elementary symmetric polynomials can be efficiently computed using [2]'s "summation algorithm".

We can further reduce the complexity of this formula by noting that rank of the $N \times N$ matrix $Q^{Y_i} = RZ^{Y_i}R$ is at most $|Y_i|$. Because $Q^{Y_i}$ only has $|Y_i|$ non-zero eigenvalues, it is the case that, for all $r$:

$$\hat{\lambda}_r e_{|Y_i|-1}^{-r}(Q^{Y_i}) = e_{|Y_i|}^N(Q^{Y_i}). \tag{B.9}$$

Recalling that the eigenvectors and eigenvalues of $Q^{Y_i}$ are denoted $\hat{V}, \hat{\Lambda}$, the computation of the singleton marginals of $q$ that are necessary for the M-step eigenvalue updates can be written as follows:

$$q(j \in J \mid Y_i) = \frac{1}{e_{|Y_i|}^N(Q^{Y_i})} \sum_{r=1}^N \hat{v}_r(j)^2 \hat{\lambda}_r e_{|Y_i|-1}^{-r}(Q^{Y_i}) = \sum_{r=1}^{|Y_i|} \hat{v}_r(j)^2. \tag{B.10}$$

This simplified formula is dominated by the $O(N^3)$ cost of the eigendecompositon required to find $\hat{V}$. This cost can be further reduced, to $O(Nk^2)$, by eigendecomposing a related matrix instead of $Q^{Y_i}$. Specifically, consider the $|Y_i| \times |Y_i|$ matrix $H^{Y_i} = V_{Y_i}R^2V_{Y_i}^\top$. Let $\tilde{V}$ and $\tilde{\Lambda}$ be the eigenvectors and eigenvalues of $H^{Y_i}$. This $\tilde{\Lambda}$ is identical to the non-zero eigenvalues of $Q^{Y_i}$, $\hat{\Lambda}$, and its eigenvectors are related as follows:

$$\hat{V} = RV_{Y_i}^\top \tilde{V} \text{diag}\left(\frac{\mathbf{1}}{\sqrt{\tilde{\boldsymbol{\lambda}}}}\right). \tag{B.11}$$

Getting $\hat{V}$ via Equation (B.11) is an $O(N|Y_i|^2)$ operation, given the eigendecomposition of $H^{Y_i}$. Since this eigendecomposition is an $O(|Y_i|^3)$ operation, it is dominated by the $O(N|Y_i|^2)$. To compute Equation (B.10) for all $j$ and requires only $O(Nk)$ time, given $\hat{V}$. Thus, letting $k = \max_i |Y_i|$, the size of the largest example set, the overall complexity of the eigenvalue updates is $O(nNk^2)$.

## C   M-Step eigenvector gradient

Recall that the M-step objective is:

$$F(V, \Lambda) = \sum_{i=1}^{n} \sum_{J} q(J \mid Y_i) \left[ \log p_K(J) + \log p_K(Y_i \mid J) \right] . \tag{C.12}$$

The $p_K(J)$ term does not depend on the eigenvectors, so we only have to be concerned with the $p_K(Y_i \mid J)$ term when computing the eigenvector derivatives. Recall that this term is defined as follows:

$$p_K(Y_i \mid J) = \mathbf{1}(|Y_i| = |J|) \det \left( \left[ V^J (V^J)^\top \right]_{Y_i} \right) . \tag{C.13}$$

Applying standard matrix derivative rules such as [4, Equation 55], the gradient of the M-step objective with respect to entry $(a, b)$ of $V$ is:

$$\frac{\partial F(V, \Lambda)}{\partial [V]_{ab}} = \sum_{i=1}^{n} \sum_{J} q(J \mid Y_i) \mathbf{1}(a \in Y_i \wedge b \in J) 2 [(W_{Y_i}^J)^{-1}]_{g_{Y_i}(a)} \cdot \boldsymbol{v}_b(Y_i) \tag{C.14}$$

where $W_{Y_i}^J = [V^J (V^J)^T]_{Y_i}$ and the subscript $g_{Y_i}(a)$ indicates the index of $a$ in $Y_i$. The $[(W_{Y_i}^J)^{-1}]_{g_{Y_i}(a)}$ indicates the corresponding row in $W_{Y_i}^J$, and $\boldsymbol{v}_b(Y_i)$ is eigenvector $b$ restricted to $Y_i$. Based on this, we can more simply express the derivative with respect to the entire $V$ matrix:

$$\frac{\partial F(V, \Lambda)}{\partial V} = \sum_{i=1}^{n} \sum_{J} 2 q(J \mid Y_i) (\dot{W}_{Y_i}^J)^{-1} \dot{V} \tag{C.15}$$

where the $\dot{V} = \mathrm{diag}(\mathbf{1}_{Y_i}) V \mathrm{diag}(\mathbf{1}_J)$ is equal to $V$ with the rows $\ell \notin Y$ and the columns $j \notin J$ zeroed. Similarly, the other half of the expression represents $(W_{Y_i}^J)^{-1}$ sorted such that $g_{Y_i}(\ell) = \ell$ and expanded with zero rows for all $\ell \notin Y_i$ and zero columns for all $\ell \notin Y_i$. The exponential-size sum over $J$ could be approximated by drawing a sufficient number of samples from $q$, but in practice that proves somewhat slow. It turns out that it is possible, by exploiting the relationship between $Z$ and $V$, to perform the first gradient step on $V$ without needing to sample $q$.

### C.1   Exact computation of the first gradient

Recall that $Z^{Y_i}$ is defined to be $U^{Y_i}(U^{Y_i})^\top$, where $U = V^\top$. The $p_K(Y_i \mid J)$ portion of the M-step objective, Equation (C.13), can be re-written in terms of $Z^{Y_i}$:

$$p_K(Y_i \mid J) = \mathbf{1}(|Y_i| = |J|) \det \left( Z_J^{Y_i} \right) . \tag{C.16}$$

Taking the gradient of the M-step objective with respect to $Z^{Y_i}$:

$$\frac{\partial F(V, \Lambda)}{\partial Z^{Y_i}} = \sum_{J} q(J \mid Y_i) (Z_J^{Y_i})^{-1} . \tag{C.17}$$

Plugging in the $k$-DPP form of $q(J \mid Y_i)$ derived in the main paper:

$$\frac{\partial F(V, \Lambda)}{\partial Z^{Y_i}} = \frac{1}{e_{|Y_i|}^N (Q^{Y_i})} \sum_{J : |J| = |Y_i|} \det(Q_J^{Y_i}) (Z_J^{Y_i})^{-1} . \tag{C.18}$$

Recall from the background section the identity used to normalize a $k$-DPP, and consider taking its derivative with respect to $Z^{Y_i}$:

$$\sum_{J : |J| = k} \det(Q_J^{Y_i}) = e_k^N(Q^{Y_i}) \underset{\text{derivative wrt } Z^{Y_i}}{\Longrightarrow} \sum_{J : |J| = k} \det(Q_J^{Y_i}) (Z_J^{Y_i})^{-1} = \frac{\partial e_k^N(Q^{Y_i})}{\partial Z^{Y_i}} \tag{C.19}$$

Note that this relationship is only true at the start of the M-step, before $V$ (and hence $Z$) undergoes any gradient updates; a gradient step for $V$ would mean that $Q^{Y_i}$, which remains fixed during the M-step, could no longer can be expressed as $R Z^{Y_i} R$. Thus, the formula we develop in this section is only valid for the first gradient step.

Plugging Equation (C.19) back into Equation (C.18):

$$\frac{\partial F(V, \Lambda)}{\partial Z^{Y_i}} = \frac{1}{e_{|Y_i|}^N(Q^{Y_i})} \frac{\partial e_{|Y_i|}^N(Q^{Y_i})}{\partial Z^{Y_i}}. \tag{C.20}$$

Multiplying this by the derivative of $Z^{Y_i}$ with respect to $V$ and summing over $i$ gives the final form of the gradient with respect to $V$. Thus, we can compute the value of the first gradient on $V$ exactly in polynomial time.

## C.2 Faster computation of the first gradient

Recall from Section B that the rank of the $N \times N$ matrix $Q^{Y_i} = RZ^{Y_i}R$ is at most $|Y_i|$ and that its non-zero eigenvalues are identical to those of the $|Y_i| \times |Y_i|$ matrix $H^{Y_i} = V_{Y_i}R^2 V_{Y_i}^\top$. Since the elementary symmetric polynomial $e_k^N$ depends only on the eigenvalues of its argument, this means $H^{Y_i}$ can substitute for $Q^{Y_i}$ in Equation (C.20), if we change variables back from $Z$ to $V$:

$$\frac{\partial F(V, \Lambda)}{\partial V} = \sum_{i=1}^{n} \frac{1}{e_{|Y_i|}^N(H^{Y_i})} \frac{\partial e_{|Y_i|}^N(H^{Y_i})}{\partial V} \tag{C.21}$$

where the $i$-th term in the sum is assumed to index into the $Y_i$ rows of the $V$ derivative. Further, because $H$ is only size $|Y_i| \times |Y_i|$:

$$e_{|Y_i|}^N(H^{Y_i}) = e_{|Y_i|}^{|Y_i|}(H^{Y_i}) = \det(H^{Y_i}). \tag{C.22}$$

Plugging this back into Equation (C.21) and applying standard matrix derivative rules:

$$\frac{\partial F(V, \Lambda)}{\partial V} = \sum_{i=1}^{n} \frac{1}{\det(H^{Y_i})} \frac{\partial \det(H^{Y_i})}{\partial V} = \sum_{i=1}^{n} 2(H^{Y_i})^{-1} V_{Y_i} R^2. \tag{C.23}$$

Thus, the initial M-step derivative with respect to $V$ can be more efficiently computed via the above equation. Specifically, the matrix $H^{Y_i}$ can be computed in time $O(N|Y_i|^2)$, since $R$ is a diagonal matrix. It can be inverted in time $O(|Y_i|^3)$, which is dominated by $O(N|Y_i|^2)$. Thus, letting $k = \max_i |Y_i|$, the size of the largest example set, the overall complexity of computing the eigenvector gradient in Equation (C.23) is $O(nNk^2)$.

# D   Baby registry experiments

Figure 1a and Figure 1b contain details, referred to in the main paper, about the baby registry dataset and the learning methods' performance on it.

| Category | N | # of Regs |
|---|---|---|
| feeding | 100 | 13300 |
| gear | 100 | 11776 |
| diaper | 100 | 11731 |
| bedding | 100 | 11459 |
| apparel | 100 | 10479 |
| bath | 100 | 10179 |
| toys | 62 | 7051 |
| health | 62 | 9839 |
| media | 58 | 4132 |
| strollers | 40 | 5175 |
| safety | 36 | 6224 |
| carseats | 34 | 5296 |
| furniture | 32 | 4965 |

(a)

| Category | Wishart | | | Moments (all data) | Moments (less data) | | |
|---|---|---|---|---|---|---|---|
| safety | (10.88) | 11.05 | (11.12) | -0.13 | (10.19) | 16.53 | (19.46) |
| furniture | (9.80) | 9.89 | (10.07) | 0.23 | (8.00) | 10.47 | (13.57) |
| carseats | (8.06) | 8.16 | (8.31) | 0.61 | (3.40) | 5.85 | (8.28) |
| strollers | (7.66) | 7.77 | (7.88) | -0.07 | (2.51) | 5.35 | (7.41) |
| health | (2.50) | 2.54 | (2.58) | 1.37 | (2.67) | 5.36 | (6.03) |
| bath | (2.50) | 2.54 | (2.59) | -0.24 | (2.22) | 3.56 | (4.23) |
| media | (2.37) | 2.42 | (2.49) | -0.17 | (0.44) | 1.93 | (2.77) |
| toys | (1.76) | 1.80 | (1.83) | 0.13 | (1.01) | 2.39 | (4.30) |
| bedding | (0.42) | 1.34 | (1.44) | 2.81 | (2.44) | 3.19 | (3.70) |
| apparel | (0.88) | 0.92 | (0.93) | 0.53 | (0.78) | 1.59 | (2.23) |
| diaper | (0.50) | 0.58 | (1.02) | -0.47 | (-0.87) | -0.19 | (1.26) |
| gear | (0.03) | 0.05 | (0.07) | 0.86 | (1.36) | 2.63 | (3.22) |
| feeding | (-0.11) | -0.09 | (-0.07) | -0.03 | (-1.32) | 0.61 | (1.22) |
| average | | 3.76 | | 0.41 | | 4.55 | |

(b)

Figure 1: (a) Size of the post-filtering ground set for each product category, and the associated number of sub-registries (subsets of $\{1, \ldots, N\}$). (b) Relative test log-likelihood differences, $100 \frac{(\text{EM}-\text{KA})}{|\text{KA}|}$, for three cases: a Wishart initialization, a moments-matching initialization, and a moments-matching initialization in a low-data setting (only $n = 2N$ examples in the training set). For the first and third settings there is some variability: in the first setting, because the starting matrix drawn from the Wishart can vary; in the third setting, because the training examples (drawn from the full training set used in the other two settings) can vary. Thus, for these two settings the numbers in parentheses give the first and third quartiles over 25 trials.