[Reviews · NeurIPS 2014]

Submitted by Assigned_Reviewer_7

This paper is concerned with learning kernel matrices to be used by determinantal point processes (DPPs). The main contribution of the paper is a reformulation of the objective that outperforms the standard approach of differentiating the logprob with respect to kernel parameters and doing projected gradient ascent (this baseline approach is called K-Ascent (KA) in the paper). The idea is to reformulate the problem within an EM framework. This works by observing that any DPP can be represented as a mixture over "elementary DPPs", where there is one elementary DPP for each subset of elements. A variational distribution is instantiated over these subsets, and the algorithm alternates between updating the distribution over these subsets, which is represented as a k-DPP, and taking single gradient steps in the M step. Results show that the EM method outperforms the KA method by quite a bit when KA is initialized naively, and by a little bit in some settings when KA is initialized more intelligently.

Overall, I find the approach interesting, and it seems like the EM method does yield some improvement. The M-step is fairly involved, and the caveats required to make it work are a bit unsatisfying, but there is something to be said for getting it to work.

Experimentally, I would have liked to have seen a bit more analysis on the effect of initialization on EM. For example, if the EM algorithm were initialized with the result of KA, would it improve upon the solution?

I'm also confused by the use of relative log likelihood as a measure to report. Log prob differences reflect ratios of probabilities. Why take ratios of log probabilities? Why not just differences in log probs? I have a hard time interpreting what these numbers mean.

Also a minor note (sorry for the crankiness!): I find some of the writing style to be annoying. I would prefer the authors don't call their own algorithm elegant, particularly when the elegance is debatable due to the caveat and approximation required in the M step. I also find it unnecessary to bold the text in the introduction.
Summary: Interesting new approach for learning kernels in DPPs. Experiments could be improved a bit, and the improvements over the basic projected gradient algorithm aren't huge, but overall a pretty good paper.

Submitted by Assigned_Reviewer_42

The paper presents and EM algorithm for learning Determinantal Point Processes. The method combines several nontrivial ingredients: eigendecomposition of the kernel, optimisation on the Stiefel manifold, representation of DPPs as mixtures of elementary DPPs, and lower-bounding the log-likelihood by Jensen 's Inequality. The experiments show favourable behaviour of the proposed approach compared to a simpler "K-Ascent" scheme.

Quality: As far as I can tell the method is well-founded, and is believable that the methods should work as stated. However,
the evaluation of the method could be improved. The experiments are quite limited and do not compare against alternative approaches for subset selection (e.g. those mentioned in the introduction), so it is not clear how much we benefit from the DPP approach.

Clarity: The paper is not particularly clearly written. write-up is somewhat sloppy and hard to follow. Not all notation is defined and sometimes definition comes after using it. The writing is also unnecessarily convoluted at times, mostly because the text is structured to describe a process how one ends up with the finally proposed approach, with side-remarks of possible alternative approaches and problems interleaved with the main thread. I would have preferred to present the developed method as clearly as possible and postpone discussion to a later stage.

Originality: For me the paper looks quite original, I am not aware of similar work.

Significance: I think the method derived in the paper presents a solid advance and a useful contribution to subset selection literature.

Details:
- p.2. line 81 the marginal kernel K is not defined before use. What kind of kernel qualifies as marginal kernel?
- p.2. l 100: V is not defined
- p.3. l 159: "minimises the change to Frobenius norm" is unclearly stated. It would be better to say "chooses the closest (in Frobenius norm) PSD matrix to Q".
- p 4. I found this section hard to understand. Rather than referring to [26] it would be nice to outline your (simpler) approach directly so that the reader is not forced to lookup the more complex original method.
- p.4 line 189: what does "weight" refer to?
- p. 7 line 324. I don't quite get the rationale in (30). Why is m_{ij} not used here? Why would simply using the empirical covariance matrix as the kernel not have worked as initialisation?
Summary: The paper puts forward a new method for learning Determinantal Point Processes for subset selection. The method is highly non-trivial and I assume it is original. The write-up is not that clear. The experiments do not consider other than DPP methods.

Submitted by Assigned_Reviewer_45

Determinantal Point Processes is a distribution over a fixed ground set that assigns higher probability to diverse sets. DPPs can be parameterized by a positive semidefinite matrix (L). Since learning L is np-hard, only partial learning of L has been discussed as prior work. Learning a scalar vector for L compromises on the diversity dimension of DPP. Previous work have resorted to restricting the parametric form of L. In this paper, the authors propose a learning method that does not restrict parameterization of L. The learning method does not require a projection step as in gradient ascent that can lead to the similarity property being compromised, and compromise on the diversity property. Using EM on eigen-values and eigen-vectors overcomes this disadvantage. They explore some optimization algorithms to solve this without needing to project values. Using projected gradient ascent requires projection of both eigen-values and eigen-vectors. Exploiting the fact that V is full-rank, they avoid the projection of eiger-vectors. Jensen’s inequality is used to lower bound the objective function and construct an EM procedure.

The learning method is useful as it preserves the diversity property of the DPP as opposed to other previous work where only the quality property has been focused upon. The step-by-step derivation of the EM procedure is insightful. Mapping the constraint to a an optimization over Stiefel manifold to eliminate the projection of eigen-vector and showing that the inverse distribution is a DPP are solid contributions. They test the learning algorithm on both synthetic datasets and a product recommendation task. The experiments support the claim in the paper –the learning method retains the quality and diversity of the DPP, by comparing it with KA. While only two of the top 10 products are replaced by KA by products that are less likely, it still brings out the contribution of the paper and supports the claim made in the beginning. The conclusion is a bit hasty not bringing forward all the contributions of this paper, but overall the paper is written cohesively. The proofs are well-written and the step-by-step reduction of optimization to EM is interesting.
Summary: The paper is written coherently and proofs are well-written. The step-by-step reduction to EM is interesting and well explained. More experiments in real-world datasets can illustrate the importance of the "diversity" produced by this algorithm when compared to others in this area.
Author Feedback
Author rebuttal: Thanks to all the reviewers for their careful consideration and
honest, helpful feedback.

R42:
- We have made the specific changes recommended with respect to
notation and organization, and will be especially careful as we
further revise to ensure all symbols are defined before they are
used. In particular, we agree that the organization of Sec 3 could
be improved and have added a paragraph outlining the structure of
the section and fronting some of the EM method's material. Other
points clarified based on your feedback include:
-- Line 81: K can be any PSD matrix with eigenvalues <= 1. This
ensures 0 <= det(K_A) <= 1, as required for marginal
probabilities.
-- Line 100: V are the eigenvectors of L and K, as in Eq (3).
-- Pg 4: There's actually no need to refer to [26] for the Stiefel
manifold algorithm; the algorithm simplifies to the single
multiplicative update shown in Eq (10).
-- Line 189: Since a DPP can be thought of as a mixture of
elementary DPPs, "weight" here refers to the mixture weights.
These are given by the p_K(J) in Eq (12).
-- Line 324: Using m_{ij} as the off-diagonal of K would not be
quite correct, as the marginal probability of the set {i,j} is
not given by K_{ij}, but rather by the 2x2 determinant
det(K_{\{i,j\}}). Setting this equal to m_{ij} results in exactly
the update given in Eq (30).
- Given that there is now a reasonable body of work demonstrating the
usefulness of DPPs as models for a wide variety of subset selection
tasks [2,3,9,12,13,14,15,16,17,18,19,20,21], we focused on learning
assuming that we already want to use a DPP, and did not make
comparisons to non-DPP-based methods. However, we acknowledge that
since this is a new task and dataset, it would be worthwhile to
include such comparisons. If there is not space for such
comparisons in the main paper body, we will include them in the
supplement.

R45:
- Although we did not find space in this version of our work for
additional real-world experiments, we acknowledge that applying our
method to additional datasets would further strengthen the paper.
Given that we observed the largest relative performance gains on the
product recommendation tasks when products exhibited strong negative
correlations, we would expect in general to see larger performance
gains on tasks where it is more important to model negative
interactions.
- We will try to make space to extend the conclusion in the final
version.

R7:
- It is a good suggestion to examine our EM algorithm initialized to
the result of KA. Although EM is more robust to initialization
because it does not have a projection step (as demonstrated in our
synthetic experiments), both EM and KA suffer from an inability to
escape from a diagonal kernel (the gradient of Eq (28) is diagonal
when K is diagonal). Since the KA algorithm often results in a
near-diagonal kernel, it is unlikely EM will be able to
substantially improve that result. However, for other failure cases
of KA, improvement may be possible. We have augmented the
discussion of these issues in the paper. To re-emphasize, KA can be
expected to exhibit poor behavior for any starting point that is
sufficiently far from optimal, whereas EM has a better chance of
escaping a bad starting configuration, provided it is not too close
to diagonal.
- The reason for scaling the difference of log-likelihoods is simply
to place the results on a scale from 0-100. For example, in Figure
2, we could have plotted just "EM - KA", but without any
normalization the relevant scale could be different for each product
category.
- We have revised our text to avoid use of ``elegant'' and bold in the
intro.